# A Spatio-Temporal Feature Trajectory Clustering Algorithm Based on Deep Learning

Xintai He, Qing Li *, Runze Wang and Kun Chen

Department of Information System Engineering, PLA Strategic Support Force Information Engineering University, Zhengzhou 450001, China; 17680327319@163.com (X.H.); wrzbleach@126.com (R.W.); xjchenkun@sina.com (K.C.)
* Correspondence: liqing0206@163.com

**Abstract:** The trajectory data of aircraft, ships, and so on, can be analyzed to obtain valuable information. Clustering is the basic technology of trajectory analysis, and the feature extraction process is one of the decisive factors for clustering performance. Trajectory features can be divided into two categories: spatial features and temporal features. In mainstream algorithms, spatial features are represented by latitude and longitude coordinates. However, such algorithms are only suitable for trajectories where spatial features are tightly coupled with latitude and longitude. When the same types of trajectories are in different latitude and longitude ranges or there are transformations such as rotation, scaling, and so on, this kind of algorithm is infeasible. Therefore, this paper proposes a spatio-temporal feature trajectory clustering algorithm based on deep learning. In this algorithm, the extraction process of the trajectory spatial shape feature is designed based on image matching technology, and the extracted spatial features are combined with the trajectory temporal features to improve the clustering performance. The experimental results on simulated and real datasets show that the algorithm can effectively extract the trajectory spatial shape features and that the clustering effect of the fused spatio-temporal feature is better than that of a single feature.

**Keywords:** feature extraction; image matching; autoencoder; trajectory clustering

## 1. Introduction

With the development of various positioning systems, such as the ADS-B (Automatic Dependent Surveillance-Broadcast) system for civil aircraft positioning [1], the AIS (Automatic Identification System) for ship positioning [2], and the most commonly used GPS (Global Positioning System), the acquisition of trajectory data has become more convenient. Trajectory data records the spatio-temporal changes of moving objects. Through the mining and analysis of the motion trajectory, the action law and behavior pattern of the moving target can be analyzed [3], and the target and motivation of the trajectory action can even be obtained [4]. This is of great significance in practical applications.

Trajectory clustering is an essential step in trajectory data analysis. It is usually used to discover the information and laws hidden in massive trajectory datasets. It plays an important role in many practical applications [5]. For example, [6] used trajectory clustering to discover waterways and establish a model to detect abnormal behavior, while [7] mined and studied driver behavior rules based on clustering. In addition, [8] observed traffic flow changes for traffic supervision based on trajectory clustering; it can be said that trajectory clustering is the basis of trajectory analysis [9].

The feature extraction process determines the clustering performance. Spatial and temporal are the essential features of trajectory data, so trajectory data is also called spatio-temporal trajectory data. The temporal features mainly reflect the correlation between the track points. The spatial features mainly reflect the overall shape of the trajectory. The current mainstream methods are biased towards the extraction of trajectory temporal features, regard trajectories as a kind of time series data, and use time series networks,

such as RNN (recurrent neural network) [10] and LSTM (long short-term memory) [11] to process trajectory data. This type of method can effectively extract the temporal features of the trajectory, but its ability to extract the spatial features is insufficient.

Therefore, in some studies, additional designs are used to extract spatial features of trajectories. For example, in [12], before using LSTM networks for classification, the distances between adjacent trajectory points were extracted as features to represent the spatial distribution of trajectories; [13] firstly counted the distribution feature vectors of the velocity, heading, acceleration, and other attributes of the trajectory as additional features of the trajectory for classification; and [14] used the shapelet classification method to extract the spatial features of trajectories. This method is similar to the traditional trajectory distance measurement algorithm, which can reflect the local spatial shape feature of the trajectory by the distance relationship between the trajectories. In [15], the trajectory spatial features were represented by extracting the slope value between two adjacent trajectory points as a feature vector.

The above improved methods achieve higher accuracy through additional trajectory space feature extraction, but they still have shortcomings. Firstly, the above improvements can only achieve the extraction of local spatial features; they cannot reflect the overall spatial features of the trajectory (for example, when trajectories need to be classified according to shape standards such as circles, straight lines, and arcs). Clustering needs to be performed according to the overall spatial characteristics of the trajectories. This overall spatial feature is mainly reflected in the shape of the trajectory, so it is called the spatial shape feature in this paper. In addition, the above methods extract spatial features from the latitude and longitude attributes, so they are only suitable for trajectories where spatial features are tightly coupled with latitude and longitude. In some applications, such as exploration, search, or some military applications, the trajectory data is not limited by the channel. Therefore, the latitude and longitude are not strictly related to the spatial features—for example, the same trajectory may be in different latitude and longitude ranges or there may be transformations such as rotation and scaling. Therefore, the methods extracting spatial features by latitude and longitude are not effective.

To summarize, the existing algorithms have insufficient ability to extract the overall spatial shape features of the trajectory. Further extraction of spatio-temporal features of trajectories is a research direction worthy of attention.

Therefore, this paper proposes a spatio-temporal feature trajectory clustering algorithm based on deep learning. In this algorithm, a method for extracting the trajectory spatial shape feature is designed first. After the trajectory is imaged, the SURF algorithm from the field of image recognition is used to extract the overall shape of the trajectory. The extracted spatial shape features are merged with the temporal features extracted by the time series autoencoder, and the fused spatio-temporal features are used for trajectory clustering. The experimental results show that the proposed algorithm can effectively extract the spatial shape features of the trajectory and show good robustness to possible rotation and scaling changes in the trajectory. The clustering effect after the fusion of the two features has also been significantly improved.

The structure of this paper is as follows. The Section 2 is the problem analysis. The Section 3 is the introduction of the algorithm. In the Section 4, algorithm performance is verified by clustering experiments. The Section 5 concludes this paper.

## 2. Problem Analysis

The trajectory can be expressed as follows:

$$\mathrm{TR}_i = \{tr_{i1}, tr_{i2}, \cdots, tr_{ik}\} \, 0 < k < N, 0 < i < M, \tag{1}$$

The trajectory point $tr_{ij}$ is composed of multi-dimensional attributes, including identity number, timestamp, position information, speed, heading, turning rate, and so on, which can be expressed as follows:

$$tr_{ij} = (\text{ID}, t_j, lat_j, lon_j, \cdots),  \tag{2}$$

The mainstream trajectory analysis method directly uses the time series network to read the temporal features in the above trajectory data. The spatial features are extracted by the latitude and longitude coordinates. This is because in some applications, the trajectory is limited by external factors, such as the channel, and the latitude and longitude coordinates of the trajectory are related to the spatial and shape characteristics of the trajectory. In fact, however, these two characteristics are different. Taking the three trajectories shown in Figure 1 as an example, trajectories 1 and 3 have high similarity from the perspective of spatial shape features. However, the mainstream algorithm uses latitude and longitude to match the trajectory points one by one according to the time series. Trajectory 1 and trajectory 2 will be classified according to the latitude and longitude distance between them. This causes the similarity of the shapes of trajectories 1 and 3 to be ignored.

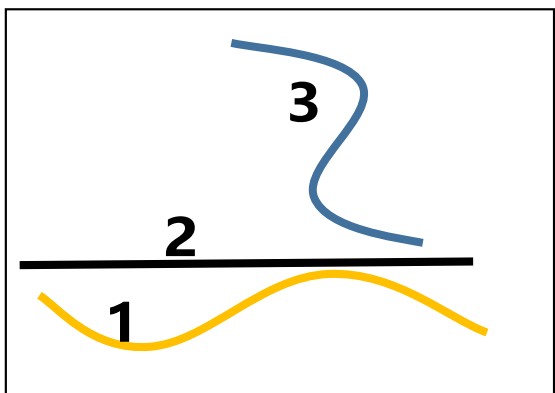

**Figure 1.** Schematic diagram of the trajectory.

Therefore, this paper will extract the spatial shape features of the trajectory separately. The first technical problem is how to extract the spatial shape features of the trajectory, and the method needs to be able to adapt to changes such as the rotation and scaling of the trajectory. In this regard, this paper introduces the image matching technology SURF (Speeded Up Robust Features) [16] method with rotation invariance to process the trajectory image and convert the result into a spatial feature vector.

The second technical problem is how to combine the extracted shape features and the temporal features. It is difficult to take full advantage of the two features by simple splicing or artificially designed weights. To this end, this paper uses an autoencoder for feature reduction and fusion to get the utmost out of the extracted two feature types.

To sum up, this paper designs a trajectory clustering algorithm based on spatio-temporal features. Its main idea is shown in Figure 2. The temporal feature is extracted by building an autoencoder with a time series network according to the conventional methods. The spatial shape feature is designed separately, including the extraction process of imaging, image matching algorithm, and feature dimension reduction processing. Finally, an autoencoder is used to fuse the two features for clustering. The specific algorithm is described in detail below.

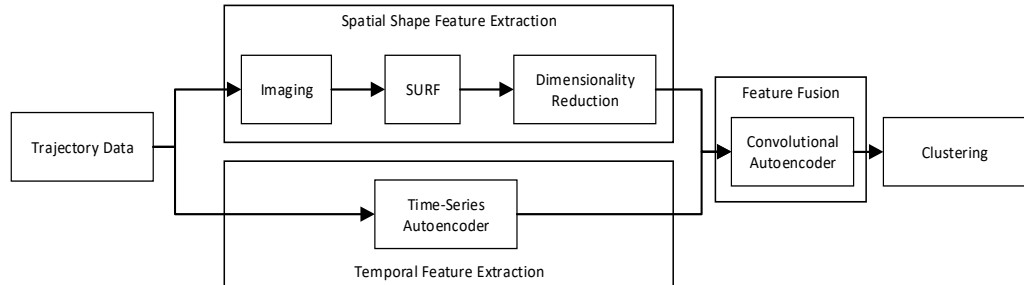

**Figure 2.** Algorithm flow.

## 3. A Spatio-Temporal Feature Trajectory Clustering Algorithm

The symbols that appear in the algorithm are described in Table 1.

**Table 1.** Symbol description.

| Symbol | Description |
|---|---|
| $TR_i$ | the *i*-th trajectory in the trajectory dataset |
| $tr_{ik}$ | the *k*-th point in the *i*-th trajectory |
| $P$ | the number of trajectory feature points |
| $P_q$ | the number of match feature points |
| $s_{ij}$ | the similarity of trajectories calculated by SURF method |
| $d_{ij}$ | the distance between the *i*-th and the *j*-th trajectory |
| $n$ | the number of track categories |
| $k$ | the number of matching samples trajectories |
| $M_i$ | the trajectory shape feature vector |
| $X_i$ | the final generated feature vector |

The remaining unmarked symbols will be further explained when they appear in the paper.

### 3.1. Image-Based Trajectory Spatial Shape Feature Extraction Algorithm

#### 3.1.1. Trajectory Imaging

In this paper, the spatial shape features of the trajectory are extracted from the trajectory image, so the trajectory needs to be imaged first. Other properties such as speed, heading, and so on, can be extracted by the temporal network. Therefore, only the position information of the trajectory points needs to be considered in the imaging process. Thus, this paper normalizes the latitude and longitude coordinates of the trajectory, removes the relative position features, and only retains the shape features of the trajectory. When processing trajectory data with elevation attributes such as airplanes, the color of the trajectory points is used in the generated image to reflect the change of this elevation attribute. The graphical result is shown in Figure 3.

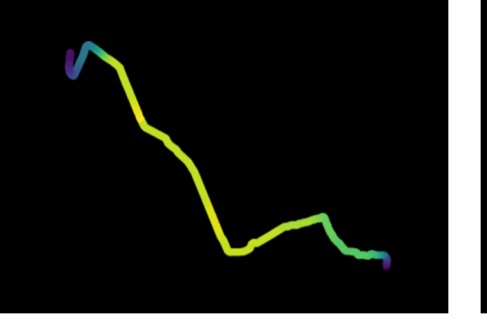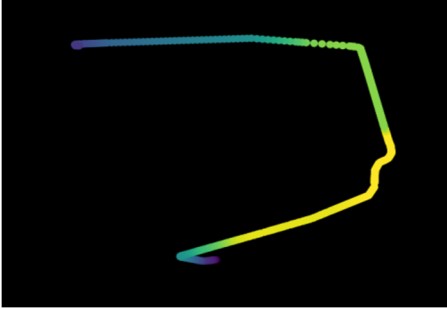

**Figure 3.** Trajectory imaging.

3.1.2. SURF Similarity Matching

The similarity of trajectory images is calculated using a classic image matching algorithm. Among the common image feature matching algorithms, the SIFT (scale-invariant feature transform) [17] algorithm is considered the most accurate algorithm, and its variant SURF algorithm processes faster. Therefore, this paper chooses the SURF algorithm to match the trajectory image.

The principle of the SURF algorithm is simply to extract the sharply changing pixel points in the image as feature points; generate feature vectors according to the position, gradient, direction, and other factors of the feature points; and use this feature point vector to match each image. This method can mark the special points, such as the corners of the trajectory, in the image and reflect the shape characteristics of the trajectory, and because the main direction of each feature point is calculated in the process of generating feature points, this algorithm has rotation invariance. Even if the image is rotated, it will not affect the matching of feature points. The matching diagram is shown in Figure 4.

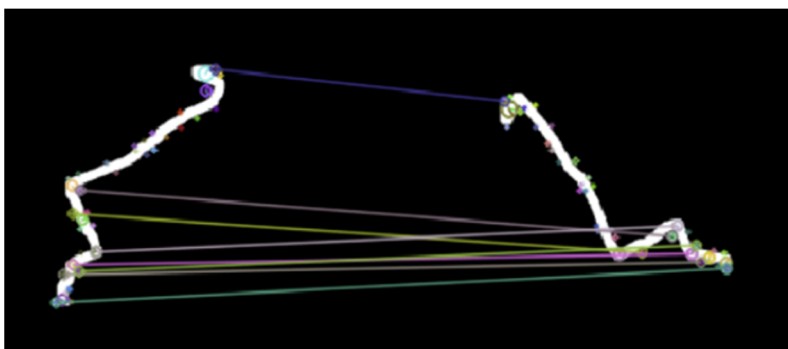

**Figure 4.** SURF for trajectory image matching.

As shown in Figure 4, the feature points of the two trajectories are matched. The number of matching points $P_q$ and the number of feature points $P$ are recorded, and the ratio of the two numbers is used to measure the similarity between these two trajectories. The formula is as follows:

$$s_{ij} = P_q / P \tag{3}$$

The larger the value of $s_{ij}$, the more similar the two trajectories are. Randomly select 100 samples in the dataset as matching targets, calculate the similarity between the trajectory and these matching samples, and use this similarity vector as the spatial shape feature vector $M_i$ of the trajectory, and we obtain the following:

$$M_i = \{s_{i1}, s_{i2}, s_{i3} \cdots s_{i100}\} \tag{4}$$

In theory, it is only necessary to match the sample with most of the possible categories to ensure that the vector reflects the overall distribution in the dataset. Assuming that there are $n$ types of data with uniform distribution in the dataset, and $k$ trajectories are extracted as matching samples, the probability that a certain type of data is not extracted is as follows:

$$p = \left(\frac{n-1}{n}\right)^k \tag{5}$$

As long as $k$ is guaranteed to have a certain size, unless there is an extreme class imbalance, the probability of one of the classes not being drawn is extremely low. In addition, in the case of extreme imbalance, the de-sampling of some minority classes has little effect on the overall similar features. In this paper, 100 trajectories are selected as matching samples, and the experimental part in Section 4 proves that the sampling process has no negative effects.

### 3.1.3. Feature Dimensionality Reduction

There are still many redundant features in the feature vector of length 100, which need to be further processed to obtain low-dimensional features for subsequent fusion with temporal features. Commonly used eigenvector dimensionality reduction algorithms include PCA (principal component analysis) [18], matrix decomposition, and so on. Autoencoders are also a commonly used method and have better dimensionality reduction performance for high-dimensional data [19]. Therefore, this paper uses an autoencoder to reduce the data dimension and learn the low-dimensional trajectory feature representation. The constructed network is shown in Figure 5.

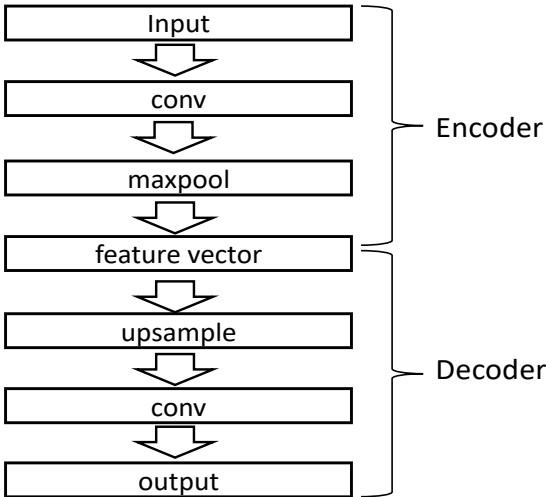

**Figure 5.** Convolutional autoencoder structure.

The convolutional layers in Figure 5 adopt the 1DCNN network. This is a convolutional network commonly used to process one-dimensional data [20]. The similarity vector extracted in this paper is also suitable for processing by this network. The main function of this network is to extract features from the input data. High-dimensional data can be transformed into corresponding low-dimensional data representations while retaining most of the critical feature information of the original data. The main function of the pooling layer is to downsample the extracted feature maps. In this paper, we choose the max pooling layer to reduce the dimensions of the feature data further. It can also improve the operating efficiency of the model and reduce the gradient disappearance and overfitting problems of the model.

The combination of convolutional and maxpooling layers can achieve the effect of data dimensionality reduction. For different input data, the size of the network needs to be changed accordingly. The ultimate goal is to maintain the same size as the features proposed by the subsequent time series network, which is convenient for subsequent feature splicing and fusion processing.

### 3.2. Spatio-Temporal Feature Trajectory Clustering Algorithm

3.2.1. Temporal Feature Extraction

An autoencoder is a neural network commonly used for feature extraction in unsupervised learning processes [21]. It is also widely used in the field of trajectory processing. For example, references [22,23] both used autoencoders to learn the features of trajectories.

Trajectories are typical time series data, and temporal features are the critical information to distinguish the type of trajectory [24]. RNN is the mainstream neural network model for processing time series data. This paper uses the RNN variant network GRU (gated recurrent unit) [25] to build a time series autoencoder. GRU is a model proposed to solve the long-term memory problem. The effect is similar to that of LSTM, but it has fewer

parameters and performs well in time series data processing. The designed autoencoder network structure is shown in Figure 6.

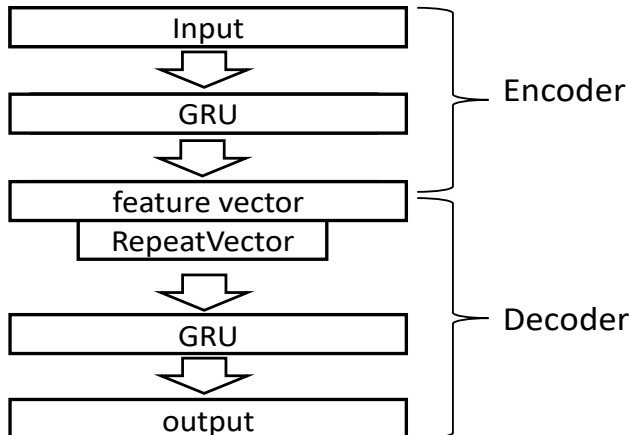

**Figure 6.** Time-series autoencoder structure.

The GRU layers in Figure 6 are used to read the trajectory points in sequence and retain the correlation between the points before and after the trajectory during processing to generate the final feature vector. The RepeatVector layer is used to regenerate the time series from the extracted feature vectors and reconstruct the input data with the symmetrical decoder network.

In the process of temporal feature extraction, the input data retains all attribute features, including latitude, longitude, speed, heading, and so on. Each attribute of the data is normalized, and zero-padding is performed to keep the length of the input data consistent. The output from the middle layer of the encoder is taken as the temporal feature of the trajectory.

### 3.2.2. Feature Fusion

Fusion of multiple features to improve data analysis results, as in [26], is common in various fields. This paper adopts a similar approach. After obtaining the two types of features, it is necessary to fuse the two for clustering. This paper continues to use the autoencoder to reduce the dimension of the data features to fuse the two features and obtain useful information.

As shown in Figure 7, the two extracted feature vectors are combined, and the (N, 2) feature vector is spliced. As in Section 3.1.3, a convolution pooling layer is used to form an autoencoder to reduce the dimension of the vector to obtain (N/2, 1) feature vectors so that the two feature vectors are fully integrated.

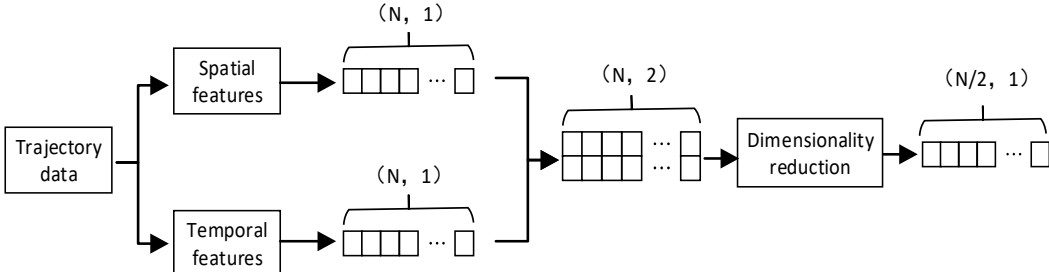

**Figure 7.** Feature fusion.

The autoencoder still uses the 1DCNN network structure, and the network structure of the corresponding size is designed according to the input and output vectors, as shown in Figure 8.

| Network Structure | | | Input | Output |
|---|---|---|---|---|
| Autoencoder | Encoder | Conv | *(N,2)* | *(N,4)* |
| | | Maxpooling | *(N,4)* | *(N/2,4)* |
| | | Dense | *(N/2,4)* | *(N/2,1)* |
| | Decoder | Dense | *(N/2,1)* | *(N/2,4)* |
| | | Upsample | *(N/2,4)* | *(N,4)* |
| | | Conv | *(N,4)* | *(N,2)* |

**Figure 8.** Fusion autoencoder network structure.

This compresses the combined features from two dimensions using convolutional and max-pooling layers. The (N, 2) features are transformed into (N/2, 1) form, and the encoder output is taken as the extracted fusion feature to fully fuse the two types of features.

### 3.2.3. Clustering

After extracting the trajectory features according to the above method, the similarity between trajectories is measured by calculating the distance between feature vectors. This paper uses Euclidean distance to measure the similarity between trajectory features [27]. Assuming that the distance between two trajectories is denoted as $d_{ij} = dist(TR_i, TR_j)$ and the extracted feature is $X_i = \{x_{i1}, x_{i2}, \cdots x_{ik}\}$, the calculation formula of the distance between the features is as follows:

$$d_{ij} = dist(X_i, X_j) = \frac{1}{k}\sum_{t=1}^{k}\sqrt{(x_{it} - x_{jt})^2} \tag{6}$$

Finally, the distance matrix is clustered based on the K-means algorithm [28], and the brief process is shown in Figure 9. Firstly, *K* cluster centers are randomly determined, and each sample point is assigned to the cluster represented by the closest cluster center point according to the similarity between each sample point and each cluster center. After all points are allocated, recalculate the new cluster center in each cluster, then re-divide all sample points according to the similarity, and repeat iteratively until the cluster center points remain stable or reach the pre-specified clustering number of runs.

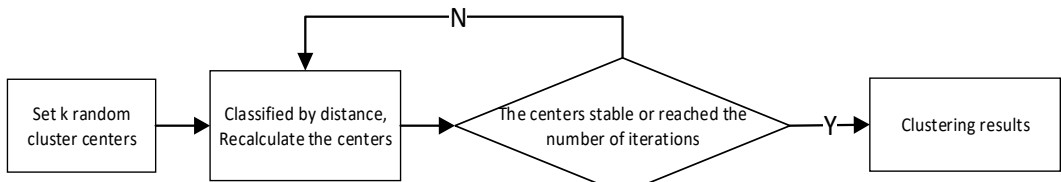

**Figure 9.** K-means clustering process.

## 4. Experiment and Analysis

### 4.1. Experimental Design

#### 4.1.1. Performance Index

The principle and improvement effect of the algorithm are verified by clustering experiments; the evaluation indicators of clustering selected are purity and KL (Kullback–Leible) divergence.

The calculation principle of purity is similar to the accuracy rate in classification, but first, the correspondence between clusters and classes needs to be assigned. The class with

the most samples in the group is taken as the representative class of the cluster. This is calculated as follows:

$$\text{Purity} = (Q, C) = \frac{1}{N} \sum_k \max_j \left| \omega_k \cap c_j \right| \tag{7}$$

In the formula, $N$ represents the total number of samples; $Q = \{\omega_1, \omega_1, \cdots \omega_j\}$ represents the clustered class; $C = \{c_1, c_1, \cdots c_j\}$ represents the correct class; $\omega_k$ represents all samples in the corresponding cluster after clustering; and $c_j$ represents the real samples of this class. The value range of purity is [0, 1]; the higher the better.

The concept of KL divergence comes from probability theory and information theory and is also known as relative entropy. It can be used to measure the difference between two distributions. The smaller the difference between the two distributions, the smaller the KL divergence. When the two distributions are consistent, the KL divergence is 0. Its calculation formula is as follows:

$$\text{KL divergence} = (p, q) = \sum_x p(x) \log \frac{p(x)}{q(x)} \tag{8}$$

In the formula, $p$ and $q$ represent the category distribution of clustering results and the distribution of actual samples, respectively. The smaller the KL divergence, the more similar the distribution of clustering results is to the actual distribution, and the better the clustering effect is.

Purity can intuitively reflect the accuracy of clustering, and KL divergence is mainly used to observe the overall clustering effect.

### 4.1.2. Data Sources

The experimental data adopted ADS-B data and GPS data.

ADS-B data is the civil aviation trajectory data recorded by the ADS-B system and downloaded from the website https://flightadsb.variflight.com, accessed on 19 July 2022. The system can periodically obtain parameters from the onboard equipment and broadcast the status information of the aircraft to other aircraft or ground stations to monitor the status of the aircraft. The information contained in the data includes flight number, time, latitude, longitude, altitude, speed, heading angle, and so on.

GPS data comes from the GeoLife dataset published by Microsoft Research. This dataset records the movement trajectories of multiple users through the GPS positioning system, including walking, bicycle, car, and other modes of transportation. It provides information such as timestamp, latitude, and longitude.

The above data is used as experimental data after preprocessing, such as screening, filtering, interpolation, smoothing, and so on. The specific data used in different experiments will be described in detail in the corresponding experimental section.

### 4.2. Verification of Trajectory Spatial Shape Feature Extraction

Firstly, the correctness of the algorithm principle is verified on the simulation dataset. We test the ability of the algorithm in this paper to read the spatial features in the trajectory image and verify the effect of introducing the extracted spatial features into the trajectory clustering process.

### 4.2.1. Simulation Datasets

Generate simulated datasets based on ADS-B data. Select the trajectories on several routes with complex spatial shape characteristics, retain only the latitude and longitude characteristics, and normalize the attributes to create a simulated trajectory dataset, as shown in Figure 10a. Only the position attribute exists in the manufactured data, and it mainly used to verify the processing ability of the algorithm for trajectory images.

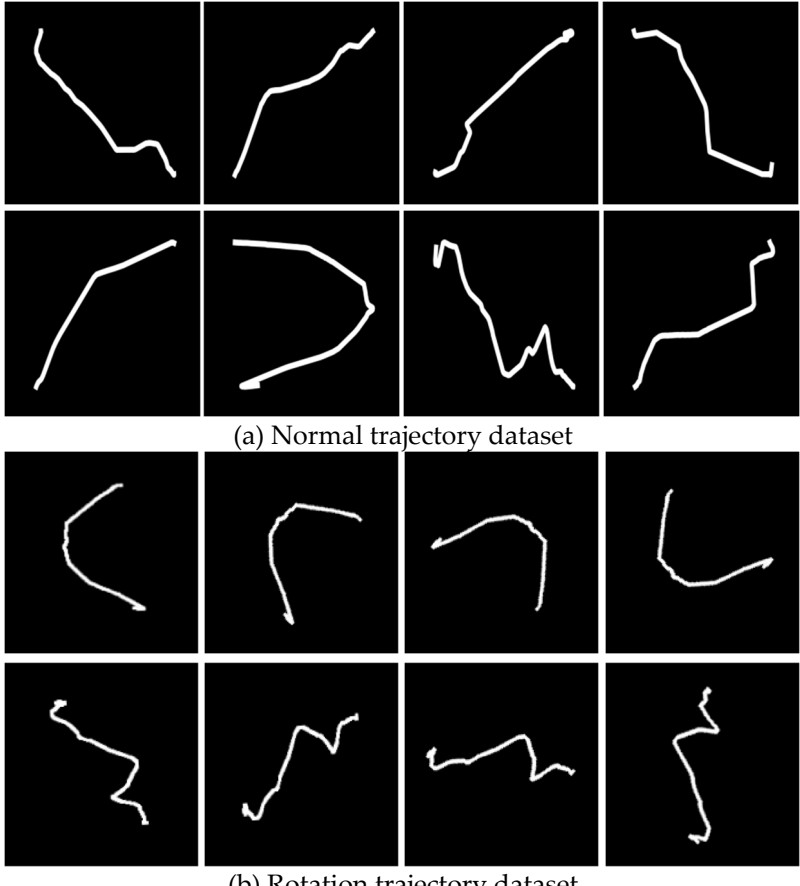

(a) Normal trajectory dataset

(b) Rotation trajectory dataset

**Figure 10.** Simulation datasets.

In addition, as introduced in the problem analysis in Section 2, trajectory shape feature extraction needs to ensure that the extraction method is robust to transformations such as rotation and scaling of the trajectory. However, since the position attribute is normalized in the imaging process, there is no need to study the scaling change of the trajectory. Therefore, based on the simulation dataset, the dataset formed by the rotation transformation of the trajectory is shown in Figure 10b, which is used to further verify the algorithm's ability to extract spatial features in the trajectory image with rotational changes.

### 4.2.2. SURF Algorithm Effect Verification

The ability of SURF algorithm to extract trajectory space features is tested. On the produced simulation dataset, the performance of SURF and the convolutional autoencoder in extracting features and clusters is compared. The temporal autoencoder model is used to process the original trajectory data as a comparison. The extraction ability of the three algorithms for trajectory shape features is compared, and the experimental results are as shown in Figure 11.

As shown in the Figure 11, in the general trajectory dataset, both the convolutional autoencoder and the temporal autoencoder can effectively read trajectory features and obtain good clustering results. However, the performance degrades rapidly when the rotation of the trajectory image appears. The SURF method is more robust to rotation changes. It shows apparent advantages in rotating datasets, which proves that the SURF method is more suitable for extracting the spatial features of the trajectory image under unsupervised conditions. Therefore, this paper chooses the SURF algorithm to extract the spatial features of the trajectory.

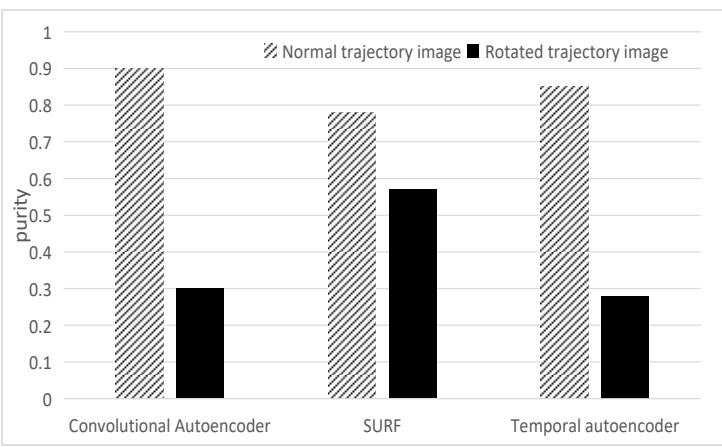

**Figure 11.** Clustering results.

4.2.3. Verification of Spatial Shape Feature Extraction Process

As introduced in Section 3.1, after using the SURF algorithm to extract the trajectory similarity in this paper, it needs to be processed by the algorithm into the trajectory space features for subsequent use. In the algorithm process, a similarity vector with a length of 100 is formed first, and then an autoencoder is used to reduce the dimensions of the vector to obtain a low-dimensional feature vector as the trajectory spatial shape feature. Sections 3.1.2 and 3.1.3 illustrate from the algorithm principle that these processing procedures do not negatively affect the spatial features extracted by the SURF method, which is experimentally verified in this section.

Observe the loss function curve during the training of the convolutional autoencoder shown in Figure 12. As the number of iterations increases, the loss value decreases rapidly and converges, and the final loss value is lower. This shows the model fits the data well, and the information loss is small in the process of feature dimension reduction.

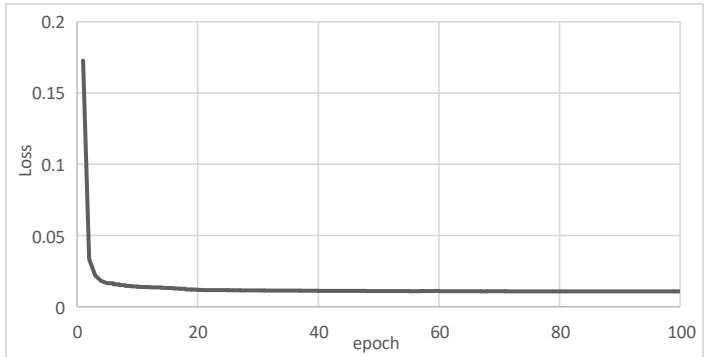

**Figure 12.** Iterative process.

Then, observe the impact of this process from the feature clustering results: clustering with the similarity matrix extracted by the SURF method, the similarity vector extracted from 100 trajectory matching, and the feature vector after dimensionality reduction processing on the simulated and the rotated dataset, respectively. The effect of the three-stage clustering is shown in Figure 13. Another method of convolutional autoencoder clustering is taken as a comparison.

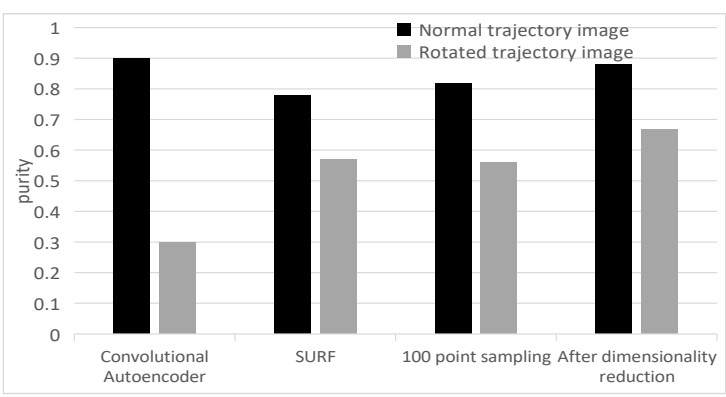

**Figure 13.** Clustering results at each stage feature.

From the experimental results, it can be seen that sampling and dimensionality reduction have no negative impact on the shape features extracted by SURF, and the shape features after corresponding processing are more suitable for trajectory clustering. The clustering effect has been significantly improved. In general simulation datasets, the same clustering effect as convolutional autoencoders can be achieved, and obvious advantages are obtained in rotated datasets. This also shows that it is appropriate to select the SURF method to extract the shape features of the trajectory in this paper.

#### 4.2.4. Verification of Feature Fusion Effect

After splicing the extracted spatio-temporal features, this paper continues to use convolutional autoencoders for feature fusion to test the effect of fusing the two features. On the simulation dataset, we compare the clustering effects of four features: time series features, spatial features, splicing features, and fusion features. The statistical results are shown in Table 2.

**Table 2.** Feature fusion effect verification.

| Clustering Features | Purity | KL Divergence |
| --- | --- | --- |
| Temporal Features | 0.73 | 0.22 |
| Spatial Shape Features | 0.87 | 0.05 |
| Feature Stitching | 0.89 | 0.06 |
| Feature Fusion | 0.93 | 0.06 |

It can be seen from the results in Table 2 that compared with the clustering results based on a single type of feature, the direct splicing and fusion of the two types of features can achieve certain improvement effects. Compared with direct splicing, the clustering effect can be improved by using the convolutional autoencoder for further dimension reduction and fusion. Observing the index of KL divergence, the overall distribution of clustering results can also remain stable during the process of splicing and fusion.

Summarizing the experiments in Section 4.2, it can be proven that the design of the algorithm in this paper, such as shape feature extraction and fusion, is correct.

#### 4.3. Algorithm Comparison

After verifying the principle and effect of the algorithm design on the simulation dataset, this section details experiments on the actual dataset to verify the actual performance and compares the algorithm effect with other classical methods.

#### 4.3.1. Datasets

One of the applications of trajectory clustering is to use the data of the aircraft takeoff to distinguish aircraft on different channels [29]. In this section, the trajectory data of 100 points after takeoff of the aircraft on three adjacent routes taking off from Shanghai

Hongqiao Airport are intercepted as the experimental dataset for clustering experiments, and the latitude and longitude coordinates of the trajectory are used to draw the trajectories, as shown in Figure 14.

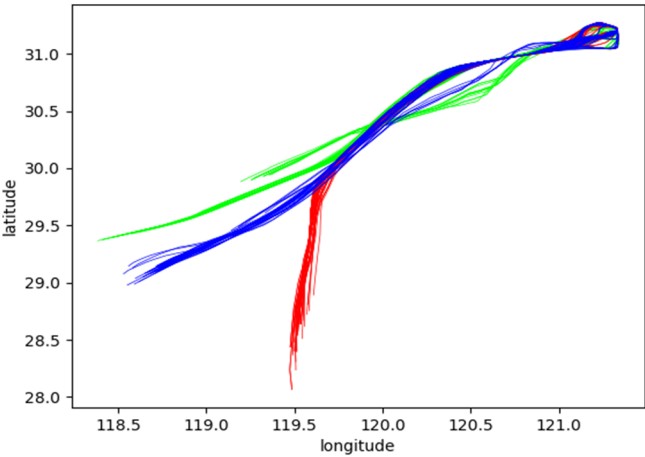

**Figure 14.** Original airplane trajectory.

In addition to position attributes, the data includes attributes such as heading and speed. In the following experiments, all attributes of the data are kept in the correlation model for temporal feature extraction. As described in the trajectory imaging method described in Section 3.1.1, only the location attributes (longitude, latitude, and elevation) are retained in the imaging process, the color changes of the trajectory points are used to reflect the trajectory elevation attributes, and the attributes are normalized. The obtained partial trajectory image is shown in Figure 15.

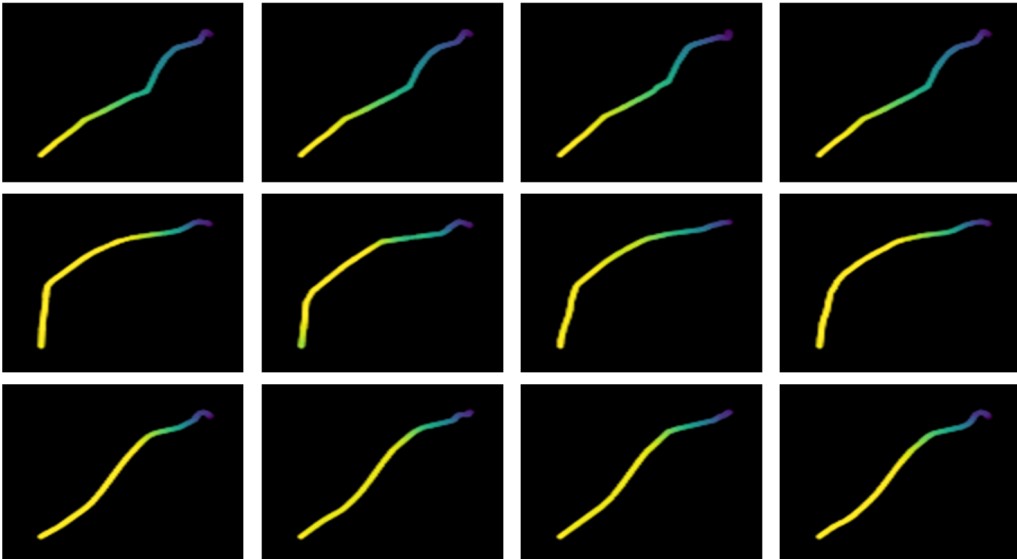

**Figure 15.** Trajectory after imaging.

In order to further verify the generality of the algorithm, this paper selects eight types of high-quality data, as shown in Figure 16, from the GeoLife dataset, and it uses the user as the label in the experimental data. The data includes three different modes of transportation: walking, cycling, and driving. As can be seen from the images, these motion patterns have their own characteristics and are more complex than civil aircraft data, so they can better test the effect of the algorithm on different trajectories.

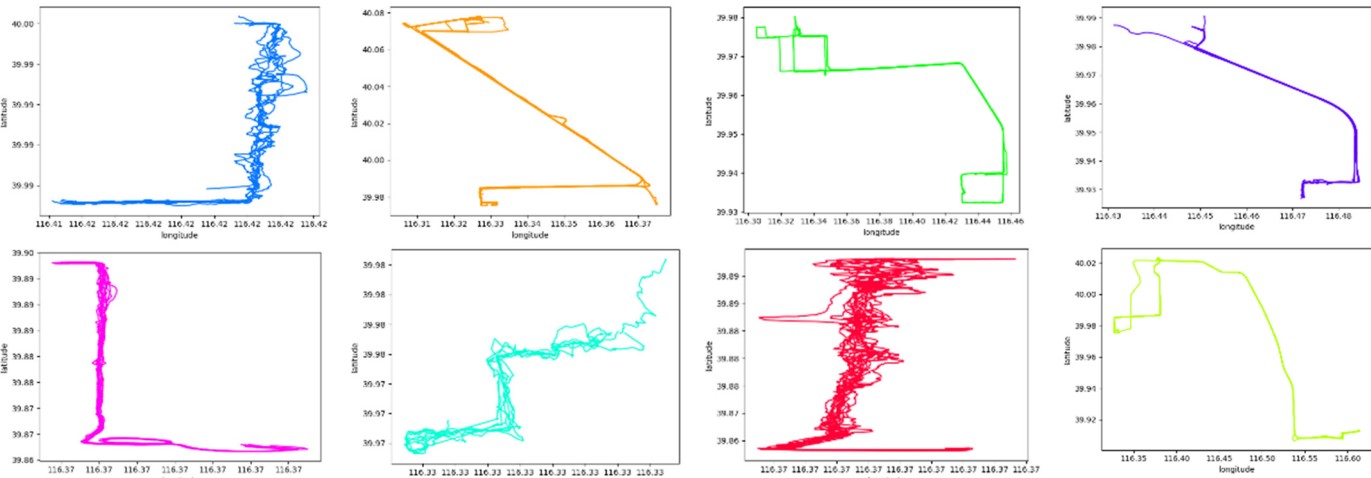

**Figure 16.** Selected data from the GeoLife dataset.

4.3.2. Experimental Results

The classical algorithms used for comparison in this section are the traditional trajectory feature extraction algorithm DTW [30] (dynamic time warping) and the MFA (multi-feature fusion autoencoder) designed in [23]. The DTW method, as a classic trajectory similarity measurement method, will not be introduced in this paper. The MFA method extracts two latent attributes, the acceleration and steering rate, from the trajectory data and combines them into three sets of attributes: latitude and longitude, velocity and acceleration, heading and steering rate. Then, it extracts them using three respective autoencoders. Compared with traditional algorithms, better robustness and clustering effect are obtained in the experiments. This method fully mines the time series attributes of trajectory data and can be used as a comparison algorithm with spatial feature mining effect.

Furthermore, the temporal autoencoder and SURF method are used alone for feature extraction and clustering to test the effect of extracting and fusing trajectory spatial features.

The datasets described in Section 4.3.1 were extracted and clustered using the above algorithm. The average statistical results of multiple experiments are shown in Table 3.

**Table 3.** Clustering results of actual datasets.

| Data | Clustering Algorithm | Purity | KL Divergence |
|---|---|---|---|
| ADS-B | SURF | 0.729 | 0.160 |
| | Temporal Autoencoder | 0.718 | 0.168 |
| | Our Algorithm | 0.882 | 0.044 |
| | DTW | 0.702 | 0.219 |
| | MFA Autoencoder | 0.790 | 0.154 |
| GeoLife | SURF | 0.719 | 0.129 |
| | Temporal Autoencoder | 0.744 | 0.149 |
| | Our Algorithm | 0.884 | 0.052 |
| | DTW | 0.791 | 0.158 |
| | MFA Autoencoder | 0.836 | 0.068 |

It can be seen from Table 3 that on the two types of experimental data, the algorithm that combines time series and space features in this paper has higher clustering purity, lower KL divergence, and better clustering results than the algorithm with a single feature.

In addition, compared with the two comparison algorithms DTW and MFA, the algorithm in this paper also has advantages. It is proven that the clustering effect of the trajectory can be effectively improved by introducing the trajectory shape feature. It also proves that our algorithm is not limited to datasets dominated by spatial features, such as the rotation datasets in Section 4.2.1. Extracting spatial shape features in other common

trajectory datasets also helps in trajectory clustering. Based on the above experiments, we can conclude that in practical applications, the algorithm in this paper can effectively extract trajectory shape features and integrate them with temporal features to improve the clustering effect.

## 5. Conclusions

Finally, a summary of the work of this paper is as follows:

(1) An image-based trajectory spatial shape feature extraction algorithm is proposed. It is used to extract the overall shape features of the trajectory and is robust to changes such as rotation and scaling of the trajectory.

(2) The extracted spatial shape features and temporal features are fused, and a trajectory clustering method based on the fusion of temporal and spatial features is proposed to get better clustering performance.

(3) The performance of the algorithm is verified by experiments on simulated datasets and actual ADS-B and GPS datasets. The experimental results show that the algorithm in this paper can effectively extract the trajectory spatial shape features and obtain better clustering performance.

The algorithm also has some shortcomings. For example, when using the SURF algorithm to extract trajectory features, the threshold in the algorithm needs to be manually designed according to the image characteristics and cannot be automatically selected. The method of adaptive threshold selection can be studied later. In addition, this paper mainly studies the extraction of trajectory spatial shape features and the improvement effect brought by the fusion of two types of features, so the network structure is not studied in depth. The next step can further optimize the network structure to enhance the feature extraction and clustering effect.

**Author Contributions:** Conceptualization, X.H.; formal analysis, X.H.; writing, X.H.; formal analysis, Q.L.; project administration, Q.L.; data curation, K.C. and R.W.; validation, K.C. and R.W. All authors have read and agreed to the published version of the manuscript.

**Funding:** This research received no external funding.

**Institutional Review Board Statement:** Not applicable.

**Informed Consent Statement:** Not applicable.

**Data Availability Statement:** The data used in this paper can be downloaded from https://flightadsb.variflight.com (accessed on 19 July 2022). or can be obtained by contacting the corresponding author.

**Conflicts of Interest:** The authors declare no conflict of interest.

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
