# Peer review of "A Spatio-Temporal Feature Trajectory Clustering Algorithm Based on Deep Learning"

_electronics, doi:10.3390/electronics11152283_

Round 1

Reviewer 1 Report

This paper proposes a deep learning-based trajectory clustering algorithm using spatio-temporal features. The algorithm proposed in this paper can be applied to clustering according to the trajectory of actual aircraft and ships. The authors extracted trajectory spatial shape features based on image matching and used them in clustering algorithms by combining the extracted spatial features and time series features. The experimental results also show relatively good results.

To improve this paper, it is necessary to increase the number of references to the literature. In order to ensure the research value, it is necessary to clarify the differentiation of this paper along with detailed references to existing studies. In this paper, only two contributions are mentioned, but I think more contributions are needed. Since the authors used only datasets for airplanes in their experiments, it has not been verified whether the model itself is a generalized model that can be utilized in various domains.

Author Response

Dear reviewer:

Thank you for your suggestion, I have made the modification according to your suggestion and marked the modified part in the paper.

The reply to your comments is as follows:

  1. According to the problem you pointed out that the paper require Moderate English changes. I have carefully revised the paper and requested my teacher to proofread and check it.
  2. Regarding the problem that you pointed out that the background introduction and references are not detailed enough, I have reorganized the introduction and problem analysis parts. And new 14 relevant references are added for introduction and clarification.
  3. In response to the problem you pointed out that the generality of the algorithm cannot be proved by using only aircraft data for experiments. I have added a clustering experiment on the GeoLife dataset released by Microsoft Research, which uses GPS data and includes multiple modes of transportation such as walking, cycling, and automobiles to verify the generalization ability of the algorithm.
  4. In response to the problem you pointed out that the contribution points of the article are not prominent enough, I have reorganized the structure of the article in the introduction and conclusion.

Thank you again for your valuable comments and hope to receive your reply as soon as possible, thank you.

Reviewer 2 Report

Dear Authors,
I have completely gone through the manuscript. Please find my detailed observations:

The Paper is based on the study of Spatio-Temporal feature trajectory clustering. Detection of targets and tracking is an important task in aircraft. 

An algorithm is proposed based on deep learning and another task proposed here is based on image matching. 

The results are more promising when compared with existing techniques.

The literature part can be added before the problem analysis (system model) or this section can remain more like the background.

This paper can be helpful for literature: An Efficient CAD based Design System for Spatial Cam Reducer

Notations can be highlighted in the paper in a table to get easy understanding

3rd section can be renamed to other title like methodology or other or proposed system model etc.

Then algorithms can be subsection in the paper.

Author Response

Dear reviewer:

Thank you for your suggestion, I have made the modification according to your suggestion and marked the modified part in the paper.

 The reply to your comments is as follows:

  1. Regarding the problem that you pointed out that the background introduction and references are not detailed enough, I have reorganized the introduction and problem analysis sections. And new 14 relevant references are added for introduction and clarification. I have benefited a lot from the literature you recommended, "An Efficient CAD based Design System for Spatial Cam Reducer", which I cite as reference [26] in the paper.
  2. Regarding your suggestion that I should create a table to introduce the symbols in the paper, I have added the corresponding table at the beginning of the Section 3.
  3. In response to your question about the revision of the title of Section 3. Corresponding to the overall algorithm of the article, I have changed the title of the third section to "A Spatio-Temporal Feature Trajectory Clustering Algorithm".

Thank you again for your valuable comments and hope to receive your reply as soon as possible, thank you.
